# S-Scheme Heterojunction Photocatalyst for Photocatalytic H$_2$O$_2$ Production: A Review

**Weili Fang and Liang Wang *** 

Key Laboratory of Marine Chemistry Theory and Technology, Ministry of Education,
College of Chemistry and Chemical Engineering, Ocean University of China, Qingdao 266100, China;
17854290741@163.com
* Correspondence: wangliangouc@ouc.edu.cn

**Abstract:** Hydrogen peroxide (H$_2$O$_2$) is a clean and mild oxidant that is receiving increasing attention. The photocatalytic H$_2$O$_2$ production process utilizes solar energy as an energy source and H$_2$O and O$_2$ as material sources, making it a safe and sustainable process. However, the high recombination rate of photogenerated carriers and the low utilization of visible light limit the photocatalytic production of H$_2$O$_2$. S-scheme heterojunctions can significantly reduce the recombination rate of photogenerated electron–hole pairs and retain a high reduction and oxidation capacity due to the presence of an internal electric field. Therefore, it is necessary to develop S-scheme heterojunction photocatalysts with simple preparation methods and high performance. After a brief introduction of the basic principles and advantages of photocatalytic H$_2$O$_2$ production and S-scheme heterojunctions, this review focuses on the design and application of S-scheme heterojunction photocatalysts in photocatalytic H$_2$O$_2$ production. This paper concludes with a challenge and prospect of the application of S-scheme heterojunction photocatalysts in photocatalytic H$_2$O$_2$ production.

**Keywords:** semiconductors; hydrogen peroxide; S-scheme heterojunction; photocatalysis





## 1. Introduction

Since its first synthesis in 1818 by Thenard [1], hydrogen peroxide (H$_2$O$_2$) has been considered a promising liquid fuel and a green oxidizer for a wide range of energy, environmental and chemical synthesis applications [2–4]. Currently, the anthraquinone (AQ) method dominates H$_2$O$_2$ production, accounting for about 95% of global H$_2$O$_2$ output [3,5]. Despite the maturity of AQ oxidation technology, it suffers from drawbacks such as high energy consumption, dangerous operation and pollution to the environment. In addition, direct synthesis of H$_2$O$_2$ using H$_2$ and O$_2$ can mitigate environmental concerns [6]. However, this method is cost-prohibitive, lacks selectivity for H$_2$O$_2$ and is prone to explosion [7]. Thus, there is a pressing need to discover an environmentally friendly and efficient H$_2$O$_2$ production method.

Solar energy is a clean and sustainable source of energy. Since Fujishima and Honda discovered the photo-assisted oxidation of water on TiO$_2$ electrodes in 1972 [8], semiconductor photocatalysis has been applied in several research fields [9–11]. Photocatalytic H$_2$O$_2$ production is a safe and green process using renewable solar energy as an energy source and resource-rich H$_2$O and O$_2$ as raw materials. In the long run, photocatalytic H$_2$O$_2$ production has great potential in environmental pollution treatment [12]. As shown in Figure 1, a great number of relevant studies have emerged in the field of photocatalytic H$_2$O$_2$ production in recent years [2,5,13–21]. However, the low visible light utilization and low solar energy conversion efficiency seriously hinder its commercial feasibility. So far, researchers have adopted various modification methods to enhance the efficiency of photocatalytic H$_2$O$_2$ production, such as doping [22,23], vacancy engineering [24], surface engineering [25], nanoparticle deposition [26] and heterojunction construction [17,27,28],

as well as combinations of two or more of these methods [29]. Thus, there is a pressing need to discover an environmentally friendly and efficient $H_2O_2$ production method.

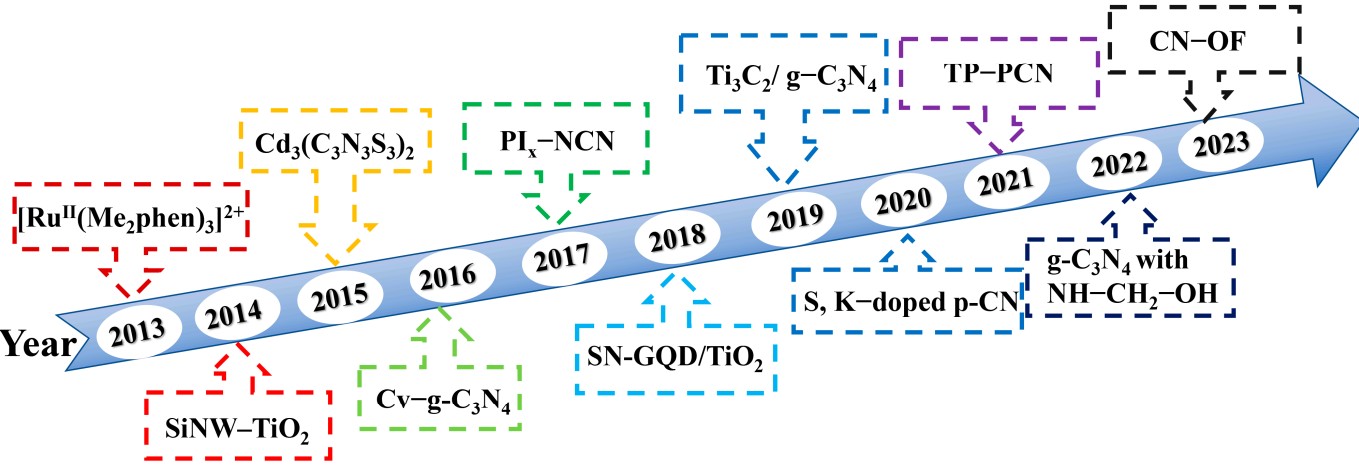

**Figure 1.** Representative photocatalysts for photocatalytic production of $H_2O_2$ in the last decade [2,5,13–21].

Mechanisms such as type-II, Z-scheme and S-scheme mechanisms are the most common in the literature used to describe the charge transfer in heterojunction structures. Although type-II heterojunctions can improve the separation efficiency of photogenerated carriers, they also sacrifice the charge of the strong redox potential, resulting in reduced redox capacity. Z-scheme photocatalysts, initially proposed by Bard in 1979, have found application in photocatalytic $H_2O_2$ production due to their effective charge separation and robust redox capabilities [30]. For instance, Cheng et al. [31] synthesized Z-scheme $Ag/ZnFe_2O_4$–$Ag$–$Ag_3PO_4$ composites for photocatalytic $H_2O_2$ production, which was generated by a continuous two-step one-electron oxygen reduction. Nevertheless, there is still some confusion about the mechanism of Z-scheme heterojunctions. Addressing the limitations inherent in type-II and Z-scheme mechanisms, Yu's team introduced the concept of S-scheme heterojunctions in 2019 [32]. The S-scheme heterojunction is composed of a reduction semiconductor and an oxidation semiconductor, which can be a p-type or n-type semiconductor. Efficient photogenerated carrier migration is achieved by the built-in electric field (IEF) at the interface of the different semiconductors, thus maintaining a high redox capacity [33,34]. In the past few years, S-scheme heterojunctions have attracted unprecedented attention because of their excellent photocatalytic activity. They are widely utilized in the fields of photocatalytic $CO_2$ reduction [35–39], photocatalytic $H_2$ production [40–44], photocatalytic $H_2O_2$ production [45] and other applications [46–49].

In this comprehensive review, we have undertaken a multi-faceted exploration of photocatalytic $H_2O_2$ production and the pivotal role played by S-scheme heterojunctions. Our journey commenced with an elucidation of the fundamental mechanism governing photocatalytic $H_2O_2$ production, followed by an in-depth analysis of the latest advancements in S-scheme heterojunctions employed within this context. Notably, recent years have witnessed remarkable progress in S-scheme heterojunction research, a modification strategy that holds immense potential for elevating photocatalyst activity and, consequently, the yield of photocatalytic $H_2O_2$ production. Our objective is to provide an in-depth reference on the $H_2O_2$ production system of S-scheme heterojunctions to stimulate new inspirations and promote the industrialization of photocatalytic $H_2O_2$ production.

## 2. Mechanism of Photocatalytic $H_2O_2$ Production Reaction

In general, the process of photocatalytic $H_2O_2$ production consists of three main steps (Figure 2). In the first step, when the absorbed photon energy of the semiconductor is greater than its band gap ($E_g$), electrons are excited and jump from the valence band (VB) to the conduction band (CB), while the hole remains in the VB, resulting in photogenerated electron–hole pairs. In the second step, the photogenerated electrons and holes separate and migrate, accompanied by the recombination of photogenerated electrons and holes, only a few of which can migrate to the surface of photocatalyst. In the last step, the electrons and holes migrating to the surface of the photocatalyst are involved in oxidation and reduction reactions, respectively. There are two main pathways for the synthesis of $H_2O_2$: oxygen reduction reaction (ORR) and water oxidation reaction (WOR).

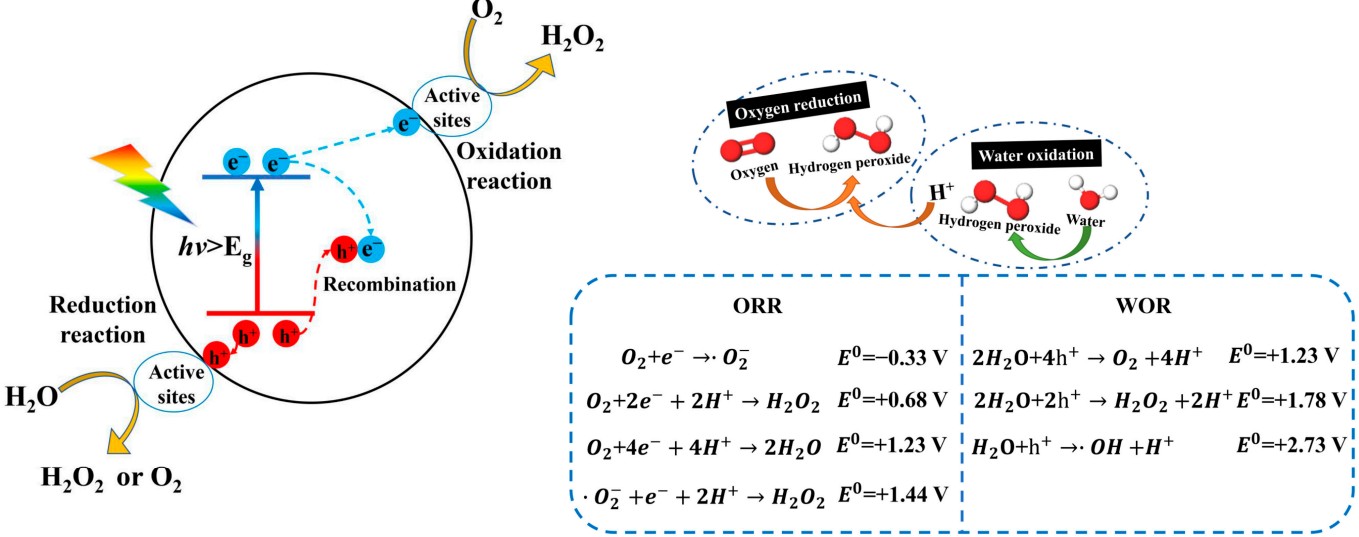

**Figure 2.** Schematic diagram of photocatalytic $H_2O_2$ production process.

The reaction potentials of photocatalytic $H_2O_2$ production are shown in Figure 2. Currently, ORR can be divided into two-step single-electron reduction ($O_2 \rightarrow \cdot O_2^- \rightarrow H_2O_2$) and direct one-step double-electron reduction ($O_2 \rightarrow H_2O_2$) routes, where the protons are mainly derived from the decomposition of $H_2O$. Since the potential of $O_2/\cdot O_2^-$ ($-0.33$ V) is much more negative than that of $O_2/H_2O_2$ (0.68 V), it requires a more negative CB position of the photocatalyst, which unavoidably increases the band gap of the photocatalyst. In general, narrow-band-gap photocatalysts are more utilized to increase their light absorption ability. Therefore, it is necessary to modify the ORR route to a one-step double-electron reaction. However, the presence of the four-electron oxygen reduction reaction makes the photocatalytic production of $H_2O_2$ less selective.

The WOR pathway is a way to synthesize $H_2O_2$ by using photogenerated holes ($h^+$) in the photocatalytic $H_2O_2$ production process. Similar to the ORR pathway, the WOR pathway can also be divided into two-electron WOR (direct two-electron and indirect two-electron) pathways and a four-electron WOR pathway. As shown in Figure 2, in the direct two-electron WOR pathway, the $h^+$ can directly oxidize $H_2O$ to $H_2O_2$ in a one-step two-electron reaction. In addition, in the indirect two-electron reaction, the $h^+$ can first oxidize $H_2O$ to hydroxyl radicals ($\cdot OH$) and then form $H_2O_2$ by coupling two $\cdot OH$. Theoretically, the direct two-electron WOR pathway requires a 1.76 V positive valence band (VB) potential of the photocatalyst, while the indirect two-electron WOR pathway requires a 2.73 V positive VB potential. The direct two-electron WOR pathway is thermodynamically more favorable but kinetically unfavorable compared to the indirect two-electron WOR pathway. Similar to the ORR pathway, the WOR pathway also results in low selectivity of $H_2O_2$ because of the competitive reaction of the four-electron WOR pathway.

In general, photocatalysts can be designed in such a way that $H_2O_2$ can be produced simultaneously by both two pathways. The dual-channel pathway integrating the ORR and WOR pathways produces $H_2O_2$ via $O_2$ and $H_2O$ without the addition of sacrificial agents and achieves 100% atomic utilization. In addition, photocatalytic $H_2O_2$ production is usually accompanied by the decomposition of $H_2O_2$. In order to improve the yield and selectivity of $H_2O_2$ in photocatalytic process, it is essential to prepare photocatalysts with suitable band gaps to provide high redox potential, high separation efficiency of photogenerated charges and excellent visible light absorption performance. To date, the performance of photocatalytic $H_2O_2$ production has been improved by such modification methods as elemental doping [19,50], morphology modulation [51], deposition of noble metals [52], vacancy engineering [53,54] and construction of heterojunctions [55–57]. Among them, the construction of heterojunctions shows excellent photocatalytic activities because it can induce the maximum separation of photogenerated carriers. Considering this, in the next section, we focus on S-scheme heterojunctions.

## 3. S-Scheme Heterojunctions

### 3.1. Mechanism of S-Scheme Heterojunctions

The separation efficiency of photogenerated carriers is an important factor for photocatalysts. In order to avoid the compounding of photogenerated carriers in a single photocatalyst, two photocatalysts were combined to enhance the photocatalytic activities. As shown in Figure 3a, in a type-II heterojunction, photogenerated carriers are generated in each of the two semiconductors under the irradiation of light. The photogenerated electrons and photogenerated holes migrate in opposite directions and aggregate on different semiconductors, thus achieving spatial separation. Although the effective separation of photogenerated carriers is achievable in type-II heterojunctions, this charge transfer reduces the redox ability of the photocatalyst. Moreover, kinetically, the presence of Coulomb repulsion inhibits this charge transfer route.

Z-scheme heterojunctions mainly include traditional Z-scheme, all-solid-state Z-scheme and direct Z-scheme heterojunctions (Figure 3b). Traditional Z-scheme and all-solid-state Z-scheme heterojunctions need to be bonded by an electron acceptor and an electron donor or a metal conductor. Thereby, electron–hole pairs with high redox capacity react with shuttling redox ion pairs or, in all-solid-state Z-scheme heterojunctions, burst each other due to greater thermodynamic driving forces [58]. Direct Z-scheme heterojunctions are derived from traditional Z-scheme and all-solid-state Z-scheme heterojunctions [59]. In a direct Z-scheme heterojunction, when two semiconductors are in contact, due to the Fermi-level difference between them, positive and negative charges collect in the interface region near the two semiconductors, resulting in an internal electric field (IEF). Photogenerated electrons are transferred from the CB of one semiconductor to the VB of the other semiconductor under the action of the IEF, as illustrated in Figure 3b. However, the term "Z-scheme heterojunction" is associated with considerable confusion, theoretical immaturity and problems. In consideration of the above disadvantages, a new charge transfer mechanism needs to be introduced to explain the charge transfer process in heterojunction photocatalysts. Thus, in 2019, Fu et al. [32] presented an S-scheme heterojunction with a similar structure to that of type-II heterojunctions which compensated for the shortcomings of Z-scheme heterojunctions [60]. As shown in Figure 3c, a S-scheme heterojunction is a coupling of an oxidizing photocatalyst (OP) and a reducing photocatalyst (RP) [61]. Like the structure of type-II heterojunctions, the OP and RP exhibit a similar interleaved structure, but the charge transfer routes between them are different. The RP with a small work function and high Fermi energy level and the OP with a large work function and low Fermi energy level form an S-scheme heterojunction by interlocking patterns. When the OP and RP are in close contact, the Fermi energy levels are bent in the interface region until the Fermi energy levels of the two photocatalysts reach equilibrium [62]. A charge accumulation layer and a charge depletion layer are formed at the interface. Energy band bending occurs in the OP and RP, which induces the recombination of electrons on the CB

in the OP and holes on the VB in the RP. As a result, the holes on the lower VB in the OP and the electrons on the higher CB in the RP are retained, favoring strong oxidation and reduction reactions, respectively [33,63]. In conclusion, by this mode formation, not only can the separation of photogenerated carriers be achieved, but the strong oxidation and reduction capabilities can also be obtained. The charge transfer path is macroscopically "step-like", so it is termed a step-scheme heterojunction.

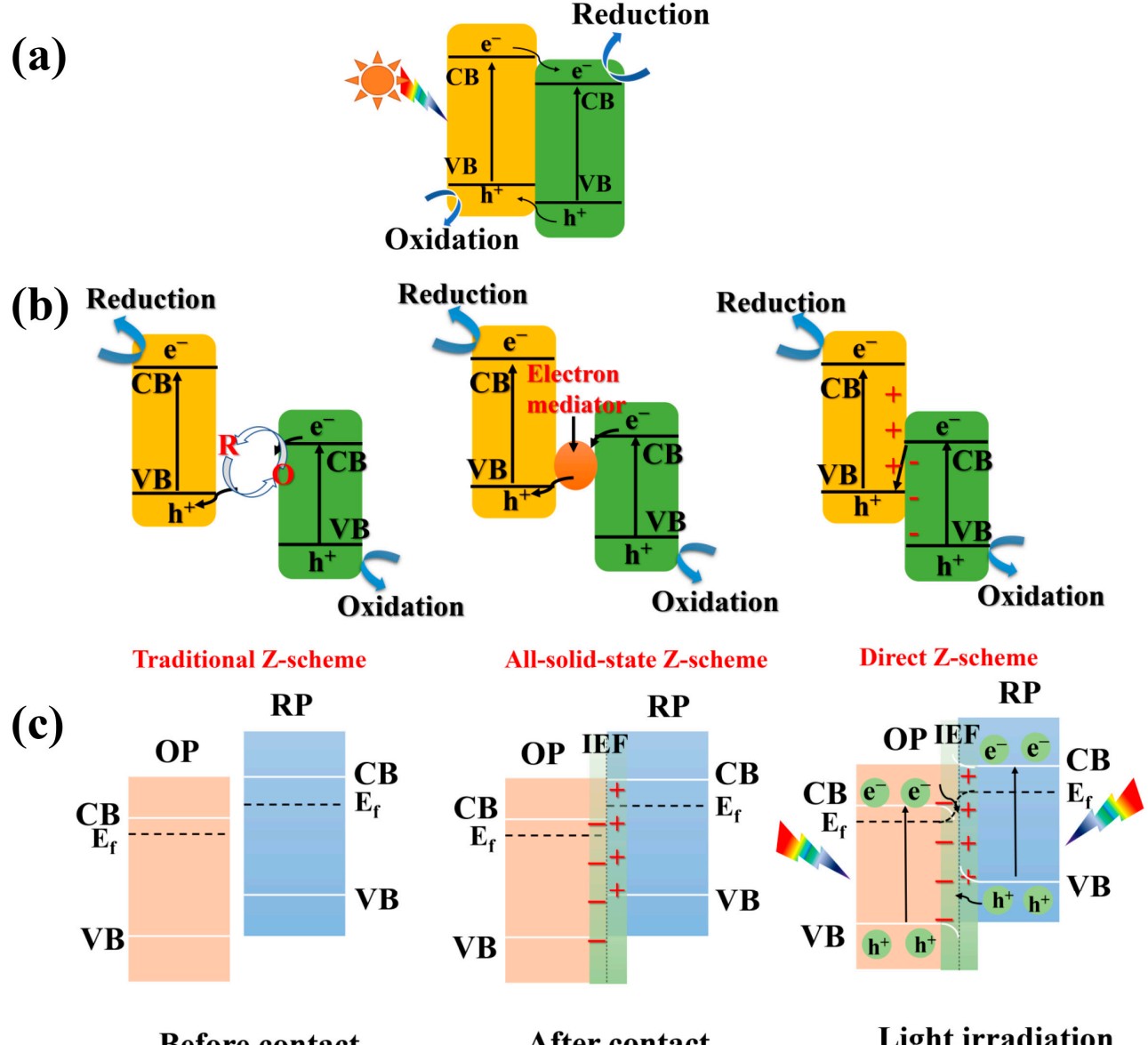

**Figure 3.** Charge transfer processes in (**a**) type-II heterojunction, (**b**) Z-scheme heterojunction, (**c**) S-scheme heterojunction: before contact; after contact; and under light irradiation.

*3.2. Characterization of S-Scheme Heterojunctions*

At the moment, the charge transfer pathway in S-scheme heterojunctions can be demonstrated by the characterization of ex situ/in situ irradiated X-ray photoelectron spectroscopy (ISIXPS), Kelvin probe force microscopy (KPFM) and electron paramagnetic resonance spectroscopy (EPR) [62]. The increase or decrease in electron density can be characterized by the shift in binding energy in the in situ XPS spectra under light conditions. The decrease in binding energy represents the increase in electron density and the atom gains electrons. Conversely, the increase in binding energy represents the decrease in

electron density and the atom loses electrons [34,64]. Thus, it can be used to determine the direction of charge transfer in heterojunction photocatalysts. For example, Yu et al. synthesized hierarchical $TiO_2$@$ZnIn_2S_4$ core–shell hollow spheres and determined the electron transfer paths by XPS. As shown in Figure 4b,c, Ti 2p and O 1s of $TiO_2$@$ZnIn_2S_4$ shifted to lower energy levels under dark conditions compared to $TiO_2$, indicating an increase in the electron density of $TiO_2$. The binding energies of Zn 2p, In 3d and S 2p of $TiO_2$@$ZnIn_2S_4$ under dark conditions were shifted to higher energy levels compared to those of $ZnIn_2S_4$ (Figure 4d–f). This indicates that electrons migrate from $ZnIn_2S_4$ to $TiO_2$ when the two photocatalysts are in contact. When light is irradiated, the electron transfer is reversed. That is, the photogenerated electrons migrate from $TiO_2$ to $ZnIn_2S_4$. This matches the charge transfer mechanism of the S-scheme heterojunction shown in Figure 4a. In addition, space charge separation in heterojunctions can be revealed by photoirradiated Kelvin probe force microscopy (KPFM) investigation. For example, Cheng et al. [65] prepared a S-scheme heterojunction by growing CdS in situ on the surface of pyrene-alt-triphenylamine conjugated polymer. Figure 5a shows an atomic force microscopy image of the photocatalyst; it can be seen that there is a surface potential difference between the two interfaces. Figure 5b,c shows the surface potential maps of the composites under dark and light conditions. As shown in Figure 5d, the surface potential difference between the PT (A) and CdS (B) is about 100 mV under dark conditions, which proves that an intrinsic electric field is formed between them pointing from the A direction to the B direction. After irradiation, the surface potential of A decreases while the surface potential of B increases. This change in surface potential proves that CdS is an electron donor in the heterojunction (Figure 5e). Furthermore, electron paramagnetic resonance (EPR) and DFT calculations can also indirectly evidence the charge transfer process [66]. EPR can be used to detect the type of radicals contained in the reaction system. Thus, to confirm that the charge transfer path of the synthesized heterojunction follows the S-scheme heterojunction photocatalyst, the presence of •OH and •$O_2$ radicals in the reaction system can be detected by EPR. It is known that the oxidation potential of OH/•OH and the reduction potential of $O_2$/•$O_2$ reach 2.73 V and –0.33 V.

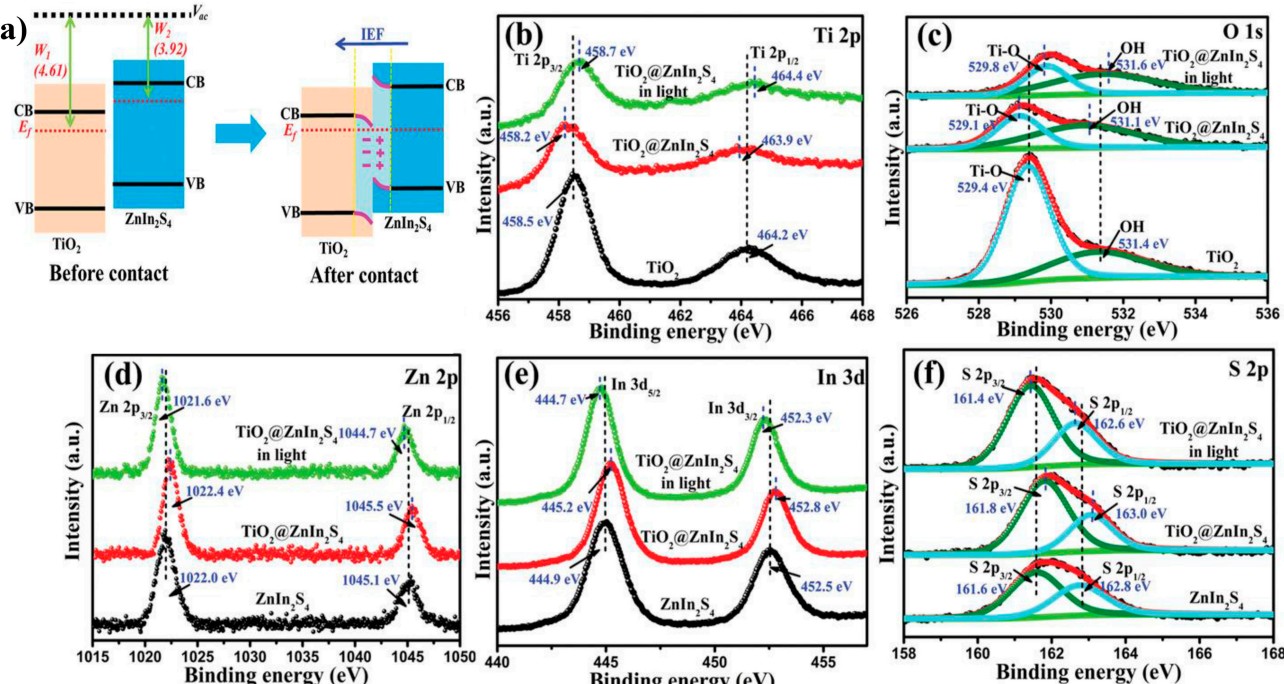

**Figure 4.** (**a**) Charge transfer processes in an S-scheme heterojunction: after contact and under light irradiation. High-resolution XPS spectra of (**b**) Ti 2p, (**c**) O 1s, (**d**) Zn 2p, (**e**) In 3d and (**f**) S 2p of photocatalysts [36].

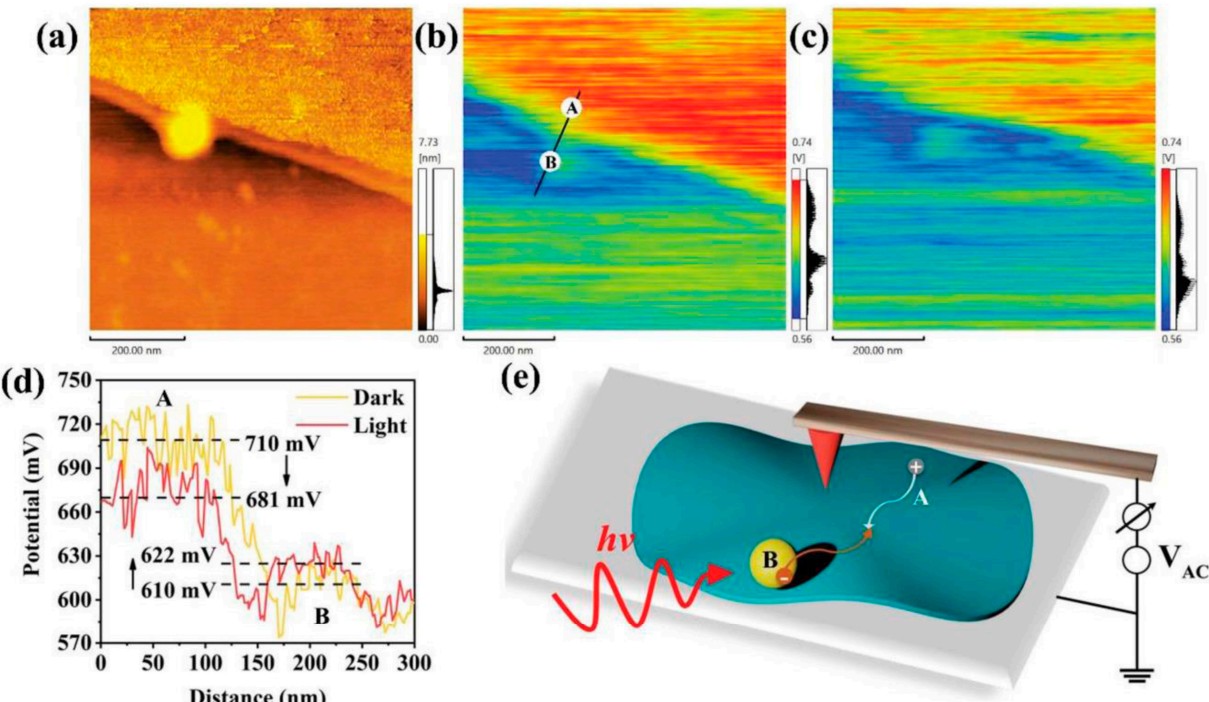

**Figure 5.** (**a**) Atomic force microscopy image of photocatalyst. Corresponding surface potential distribution of photocatalyst (**b**) in dark and (**c**) under light irradiation. (**d**) The line-scanning surface potential from point A to B. (**e**) The schematic illustration of photoirradiation KPFM [65]. (point A: PT; point B: CdS)

### 3.3. Synthesis Method

Presently, various methods to synthesize S-scheme heterojunctions exist, such as the hydrothermal/solvothermal method [67–69], sol–gel electrostatic spinning method [70,71], self-assembly method [32,72,73] and co-precipitation method [74,75]. For example, Li et al. [76] synthesized a novel S-scheme $TiO_2$/$ZnIn_2S_4$ heterojunction photocatalyst by the hydrothermal method and evaluated its photocatalytic performance by photocatalytic $H_2$ production. $TiO_2$ nanofibers are dispersed in an aqueous ethanol solution containing $Zn^{2+}$ and $In^{3+}$, which are anchored to the surface of $TiO_2$ nanofibers by Coulomb electrostatic interactions, while an S source is added. $TiO_2$/$ZnIn_2S_4$ heterojunctions are obtained by hydrothermal method. It was found the S-scheme mechanism of photogenerated charge transfer made $TiO_2$/$ZnIn_2S_4$ exhibit the highest $H_2$ production activity with a $H_2$ production rate of 6.03 mmol·$g^{-1}$·$h^{-1}$.

## 4. $H_2O_2$ Production by S-Scheme Heterojunction Photocatalysts

$H_2O_2$ production by photocatalysis is a safe, sustainable and green process because it requires only water and oxygen from the air as raw materials and sunlight as an energy source [77–79]. In S-scheme heterojunctions, the Fermi energy level difference between semiconductors induces the formation of an intrinsic electric field and energy band bending, which promotes the effective migration and separation of photogenerated electrons and holes. This advantage of S-scheme heterojunctions makes them promising for photocatalytic $H_2O_2$ production. This review focuses on the application of S-scheme heterojunctions in photocatalytic $H_2O_2$ production.

### 4.1. Photocatalytic $H_2O_2$ Production

As described in Section 2, the two main pathways for photocatalytic $H_2O_2$ production are the ORR and WOR pathways. Photocatalytic reactions mainly include light absorption, migration and separation of photogenerated charges and redox reactions on surfaces. The most important prerequisite for photocatalytic $H_2O_2$ production is to satisfy the reaction

potential of ORR and WOR pathways. Thus, the band gap position of the photocatalyst is of critical importance in $H_2O_2$ production. S-scheme heterojunctions have significant advantages in photocatalytic $H_2O_2$ production because of effective separation of photogenerated carriers and enhanced redox capacity. The oxygen reduction pathway is the most popular photocatalytic $H_2O_2$ production pathway. For example, Jiang et al. [80] synthesized S-scheme $ZnO/WO_3$ heterojunction photocatalysts for photocatalytic $H_2O_2$ production by hydrothermal and calcination methods. FESEM and TEM images show that $ZnO/WO_3$ exhibits a hierarchical microsphere structure (Figure 6a,b). The prepared $ZnO/WO_3$ heterojunctions showed superior photocatalytic activity compared to the single component. When the volume of $WO_3$ was 30%, ZW30 exhibited an $H_2O_2$ yield of 6788 $\mu mol \cdot L^{-1} \cdot h^{-1}$. In addition, cyclic tests revealed good stability of ZW30, with a small decrease in $H_2O_2$ yield after four cycles. Figure 6c depicts the mechanism of $ZnO/WO_3$ for photocatalytic $H_2O_2$ production. The process is based on a direct $2e^-$ ORR pathway, accompanied by indirect $2e^-$ ORR pathway. The characterization and experimental results demonstrate the formation of a $ZnO/WO_3$ S-scheme heterojunction with a structure capable of providing more reducing electrons, thus enhancing the driving force of $H_2O_2$ production by ORR. In another work, Lai et al. [81] developed a $CdS/K_2Ta_2O_6$ S-scheme heterojunction by a two-step hydrothermal method, which exhibits excellent photocatalytic $H_2O_2$ production activity without using any sacrificial agent and additional $O_2$. The SEM image shows that the $CdS/K_2Ta_2O_6$ composite exhibits a flower-like structure (Figure 7a). In situ irradiated XPS, EPR and DFT calculations were used to propose the mechanism of an S-scheme heterojunction for $H_2O_2$ production (Figure 7b). The simultaneous presence of WOR and ORR pathways enables efficient utilization of the redox system. All the above studies provide insights into the design of S-scheme heterojunction photocatalysts for efficient photocatalytic $H_2O_2$ production. In recent years, there have been a number of S-scheme heterojunctions applied in photocatalytic $H_2O_2$ production. Table 1 presents the studies of S-scheme heterojunctions for photocatalytic $H_2O_2$ production.

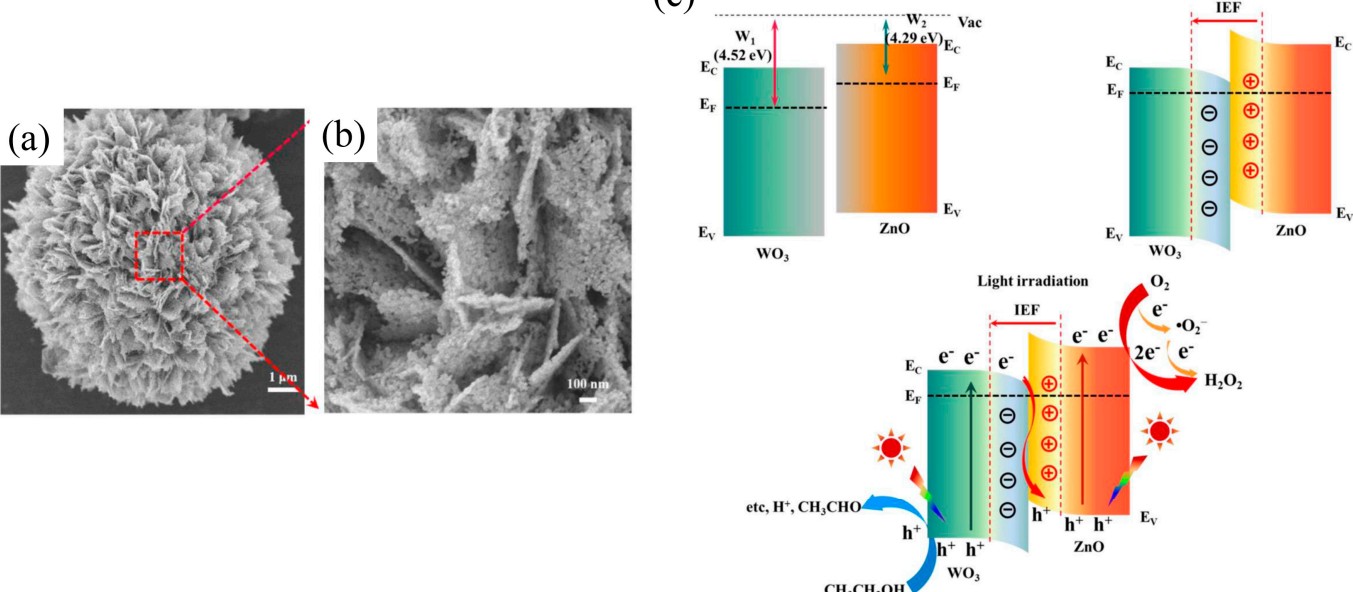

**Figure 6.** (**a**,**b**) FESEM images of $ZnO/WO_3$. (**c**) Photocatalytic $H_2O_2$ production mechanism of $ZnO/WO_3$ photocatalyst [80].

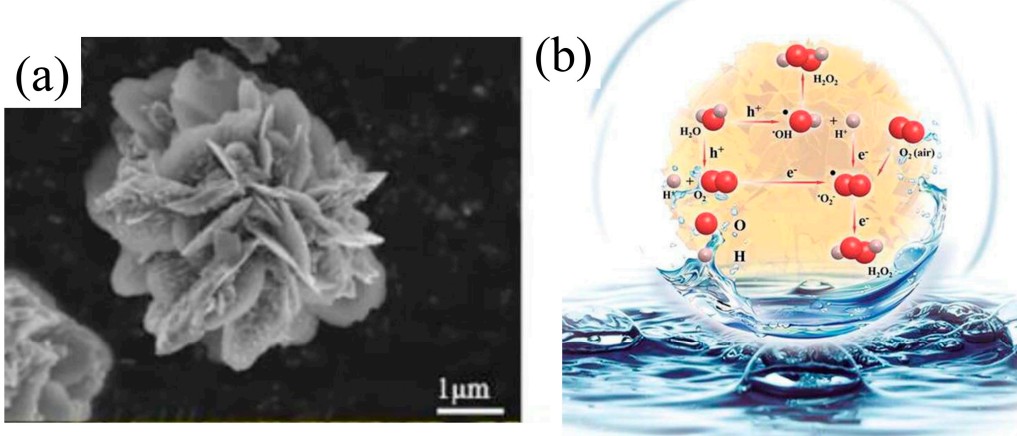

**Figure 7.** (**a**) Representative SEM image of $CdS/K_2Ta_2O_6$ photocatalyst. (**b**) Photocatalytic $H_2O_2$ production mechanism of $CdS/K_2Ta_2O_6$ photocatalyst [81].

**Table 1.** Studies of S-scheme heterojunctions for photocatalytic $H_2O_2$ production.

| Photocatalyst | Morphology | Light Source | Reaction Solution | Pathway | Concentration of Photocatalyst/$g \cdot L^{-1}$ | $H_2O_2$ Yield | Ref. |
|---|---|---|---|---|---|---|---|
| $ZnO/WO_3$ | Hierarchical microsphere structure | 300 W Xe lamp | 50 mL of 10 vol% ethanol | Direct $2e^-$ ORR and indirect $2e^-$ ORR pathways | 1.0 | 6788 $\mu mol \cdot L^{-1} \cdot h^{-1}$ | [80] |
| $CdS/K_2Ta_2O_6$ | Flower−like structure | 300 W Xe lamp ($\lambda > 420$ nm) | Ultra−pure water | $2e^-$ ORR and WOR pathways | 0.6 | 160.89 $\mu mol \cdot L^{-1} \cdot h^{-1}$; 346.31 $\mu mol \cdot L^{-1} \cdot h^{-1}$ with saturated $O_2$ | [81] |
| $ZnO/g\text{-}C_3N_4$ | ZnO NPs dispersed on the CN nanosheet | 300 W Xe lamp ($\lambda > 350$ nm) | 50 mL of 10 vol% ethanol | ORR pathway | 0.4 | 1544 $\mu mol \cdot L^{-1} \cdot h^{-1}$ | [82] |
| $TiO_2/In_2S_3$ | Core−shell structure | 300 W Xe arc lamp | 40 mL of 10 vol% ethanol | Indirect $2e^-$ ORR pathway | 0.5 | 376 $\mu mol \cdot L^{-1} \cdot h^{-1}$ | [45] |
| $C_3N_4/PDA$ | Nanosheet | 300 W Xe arc lamp ($\lambda > 350$ nm) | 50 mL of 20 vol% ethanol | Indirect $2e^-$ ORR pathway | 0.4 | 3801.25 $\mu mol \cdot g^{-1} \cdot h^{-1}$ | [83] |
| $ZnO/COF$ (TpPa−Cl) | ZnO nanoparticles distributed on the surface of TpPa−Cl | 300 W Xe lamp | 10 vol% ethanol | Indirect $2e^-$ ORR pathway | 0.5 | 2443 $\mu mol \cdot g^{-1} \cdot h^{-1}$ | [84] |
| $TiO_2/PDA$ | Inverse opals | 300 W Xe arc lamp | 40 mL of 10 vol% ethanol | ORR pathway | 0.5 | ~2.2 $mmol \cdot g^{-1} \cdot h^{-1}$ | [85] |
| $In_2O_3/ZnIn_2S_4$ | Ordered hollow structure | 250 W Xe lamp ($\lambda > 420$ nm) | 50 mL of 5 vol% ethanol | ORR pathway | 0.4 | 5716 $\mu mol \cdot g^{-1} \cdot h^{-1}$ | [86] |
| Sv−ZIS/CN | Three−dimensional flower-like structure and agaric shaped with a microporous structure | 300 W Xe lamp ($\lambda > 420$ nm) | 50 mL of 10 vol% isopropanol | Direct $2e^-$ ORR and indirect $2e^-$ ORR pathways | 0.4 | 1310.18 $\mu mol \cdot L^{-1} \cdot h^{-1}$ | [87] |
| $Bi_2S_3@CdS@RGO$ | Flaky RGO is wrapped onto the CdS nanoparticles and $Bi_2S_3$ rod−aggregate morphology | 300 W Xe lamp ($\lambda > 420$ nm) | 50 mL of 10 vol% isopropanol | Indirect $2e^-$ ORR pathway | 1.0 | 212.82 $\mu mol \cdot L^{-1}$ within 180 min | [88] |
| ZnO@PDA | Inverse Opal | 300 W Xe arc lamp | 50 mL of 4 vol% glycol | Direct $2e^-$ ORR and indirect $2e^-$ ORR pathways | 0.4 | 1011.4 $\mu mol \cdot L^{-1} \cdot h^{-1}$ | [89] |
| $S\text{-}pCN/WO_{2.72}$ | Uniform porous sheet−like two−dimensional structure | 300 W Xe lamp ($\lambda > 420$ nm) | 100 mL water | Direct $2e^-$ ORR and indirect $2e^-$ ORR pathways | 1.0 | 90 $\mu mol \cdot L^{-1}$ within 180 min | [90] |
| $TiO_2@RF$ | Core−shell structure | 300 W Xe lamp | 15 mL water | $2e^-$ ORR pathway | ~0.67 | 66.6 $mmol \cdot g^{-1} \cdot h^{-1}$ | [91] |
| sulfur-doped $g\text{-}C_3N_4/TiO_2$ | Well-ordered macroporous framework | 300 W Xe lamp | 50 mL water | $2e^-$ ORR and WOR pathways | 0.2 | 2128 $\mu mol \cdot g^{-1} \cdot h^{-1}$ | [92] |

### 4.2. Water Splitting

$H_2O_2$ can also be used as a valuable by-product of photocatalytic overall water splitting to produce $H_2$. Photocatalytic $H_2$ production from overall water splitting has been a hot research problem; however, it has the disadvantages of slow kinetics and difficult product separation. The production of $H_2$ and $H_2O_2$ from pure water by a two-electron photocatalytic mechanism solves the above problems due to a lower reaction potential than that of the four-electron reaction [93,94]. Two-electron overall water splitting thermodynamically requires a stronger oxidation capacity of the photocatalyst. S-scheme heterojunctions have a strong redox capacity because of their unique step-scheme charge transfer mechanism. For instance, Meng et al. [95] successfully synthesized a g-$C_3N_4$/$CoTiO_3$ S-scheme heterojunction photocatalyst and applied it in photocatalytic overall water splitting for $H_2$ production under visible light. The $H_2$ production efficiency was significantly improved without sacrificial agents, while the presence of $H_2O_2$ was detected in the photocatalytic process. Based on the results of EPR and DFT calculations, the possible reaction mechanism of the photocatalyst is shown in Figure 8. The difference in the Fermi energy levels of CN and $CoTiO_3$ results in the formation of an intrinsic electric field (IEF) at the contact surface of the two photocatalysts. As a result, energy band bending also occurs in the interface region, forming an S-scheme heterojunction. This means of charge transfer promotes the migration and separation of photogenerated carriers and preserves the strong redox ability of the system, which is beneficial in enhancing the efficiency of photocatalytic overall water splitting.

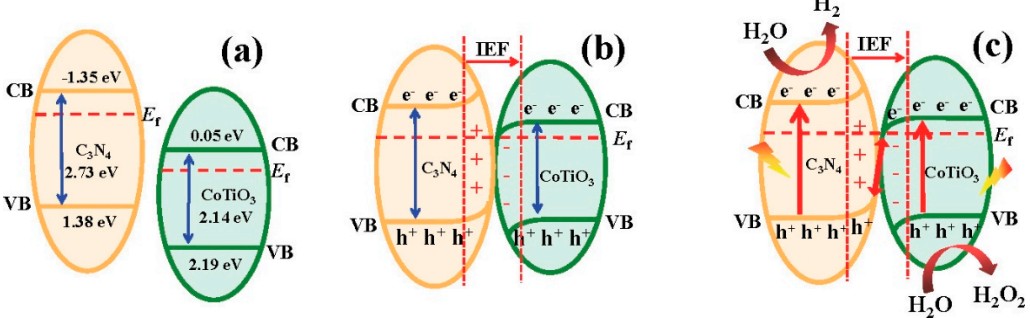

**Figure 8.** Schematic of the energy levels of CN and $CoTiO_3$ (**a**) before contact, (**b**) after contact and (**c**) under irradiation in S-scheme reaction mechanism of CCT−1.5 [95].

### 4.3. Coupling of $H_2O_2$ Production and Organic Synthesis

S-scheme heterojunctions can maximize the redox ability of photocatalysts, effectively utilizing photogenerated electrons and holes and, therefore, having the ability to simultaneously achieve the reduction of $O_2$ to $H_2O_2$ and the oxidation of organics [96]. For instance, He et al. [55] synthesized floatable S-scheme $TiO_2$/$Bi_2O_3$ photocatalysts by immobilizing hydrophobic $TiO_2$ and $Bi_2O_3$ on lightweight polystyrene (PS) spheres by hydrothermal and photodeposition methods. The photocatalysts showed significant $H_2O_2$ yields and were able to oxidize furfuryl alcohol (FFA) to furoic acid (FA). The mechanism of the photocatalytic reaction was revealed by in situ DRIFT spectroscopy and DFT calculations (Figure 9a,b). In addition, the floatable photocatalyst is able to be in closer contact with $O_2$ compared to conventional biphasic photocatalytic systems, solving the problem of slow transport of gas reactants from suspended photocatalysts (Figure 9c). Moreover, floatable photocatalysts are less prone to agglomeration, easy to recover and can be recycled. The floatable S-scheme heterojunction photocatalyst not only improves the efficiency of photocatalytic reactions but also provides a new idea for efficient multiphase catalysis. In addition, recently, Yu et al. successfully prepared S-scheme $TiO_2$@BTTA photocatalysts by synthesizing COF (BTTA) via Schiff-base condensation and by encapsulating $TiO_2$ NF with BTTA COF. The heterojunction photocatalysts show high $H_2O_2$ production activity and furoic alcohol (FAL) oxidation activity, with a $H_2O_2$ production rate of 740 μmol·$L^{-1}$·$h^{-1}$ and a FAL conversion of 96%.

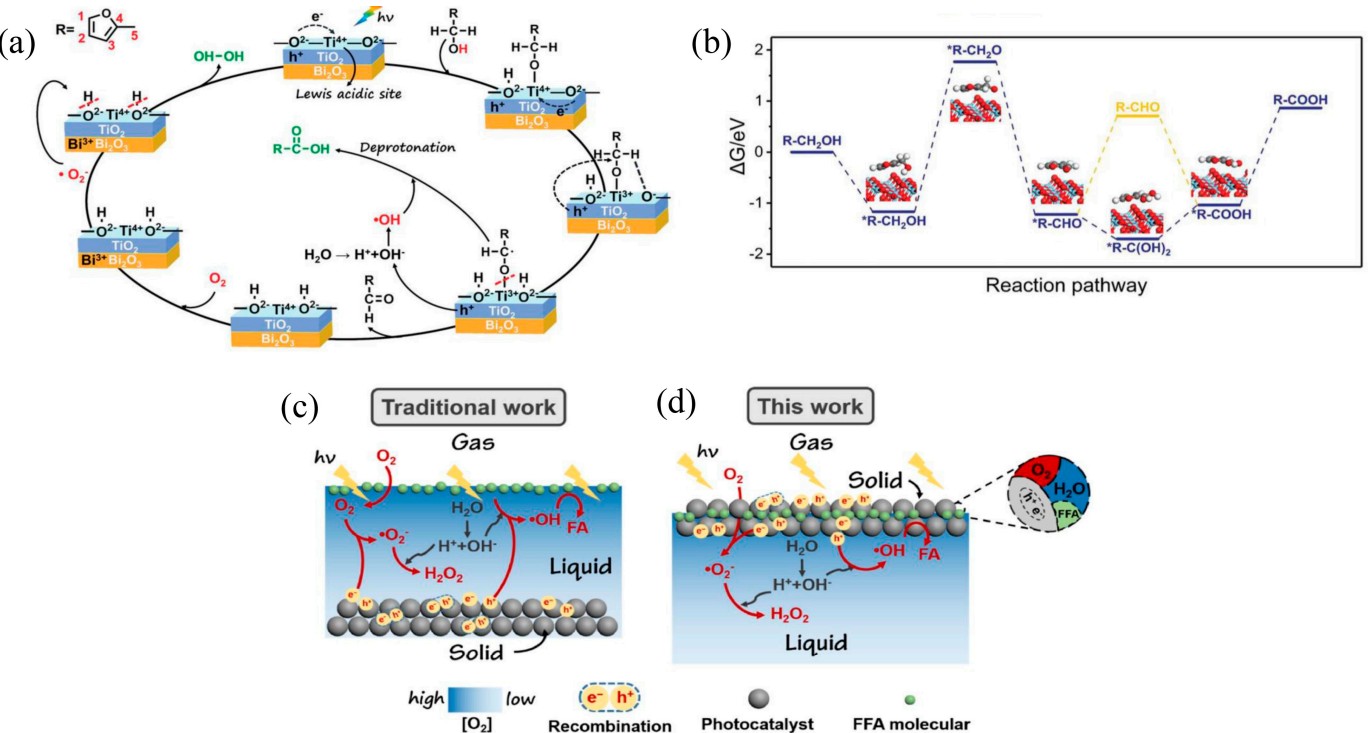

**Figure 9.** (**a**) The mechanism of photocatalytic FFA oxidation coupling with $H_2O_2$ production on surface of photocatalyst. (**b**) Free energy diagrams of FFA oxidation steps on active sites of $TiO_2$. Schematic diagram of $O_2$ supply for photocatalyst in (**c**) biphase and (**d**) triphase system [55].

*4.4. Pollutant Degradation with In Situ $H_2O_2$ Production*

    $H_2O_2$ is usually used in the degradation of contaminants due to its oxidizing ability to improve photocatalytic degradation efficiency. In general, the reactive oxygen species (ROS) used for photocatalytic degradation are mainly $H_2O_2$, $\cdot O_2^-$ and $\cdot OH$. $H_2O_2$ is the only stable molecule among them and has a longer lifetime than other active radicals. In situ $H_2O_2$ production to enhance the degradation of contaminants in photocatalytic processes has proven to be an effective strategy. Recently, S-scheme heterojunction photocatalysts have also been developed for this application (Table 2). Li et al. [97] synthesized a novel layered BP/BiOBr S-scheme heterojunction by self-assembling BiOBr nanosheets on the surface of BP nanosheets by liquid-phase sonication combined with solvothermal methods. The composite exhibited excellent photocatalytic degradation activity of tetracycline (TC) under visible light, which was 7.8 times higher than that of pure BiOBr. The increased activity was attributed to the structure of S-scheme heterojunctions retaining a high redox capacity. The active groups during the experiment were tested by ESR characterization, as shown in Figure 10a,b. After illumination, the signals of both $\cdot O_2^-$ and $\cdot OH$ groups were detected, but the signals of $\cdot O_2^-$ groups became lower with the increase in illumination time, indicating that some $\cdot O_2^-$ and $H^+$ formed $H_2O_2$. The results indicate that the main active substances of TC mineralization are in situ generated $H_2O_2$ and $\cdot OH$. Based on the above results, the photocatalytic mechanism of the S-scheme heterojunction is proposed as shown in Figure 10c.

**Table 2.** Studies of S-scheme heterojunction photocatalytic $H_2O_2$ production coupled with pollutant degradation.

| Photocatalyst | Morphology | Contaminant or Organics | Light Source | Reaction System | Concentration of Photocatalyst/g·L$^{-1}$ | $H_2O_2$ Yield | Degradation Efficiency | Ref. |
|---|---|---|---|---|---|---|---|---|
| PDI−Urea/BiOBr | BiOBr nanospheres dispersed on the PDI−Urea lamellar layer | Ofloxacin (OFLO), tetracycline (TC) | 300 W Xe lamp (λ > 420 nm) | 50 mL of TC (50 mg/L) and OFLO (10 mg/L) | 1.0 | 71 µmol·L$^{-1}$·h$^{-1}$ after 3 h irradiation | 93%(~65%) photocatalytic degradation rate for OFLO (TC) after 150 (90) min | [98] |
| BP/BiOBr | Two−dimensional structure | Tetracycline (TC) | 300 W Xe arc lamp (λ > 420 nm) | 100 mL of TC (50 mg/L) | 1.0 | 1.62 µmol·L$^{-1}$·min$^{-1}$ | ~85% photocatalytic degradation rate for TC after 90 min | [97] |
| Graphitic−C$_3$N$_4$/ZnCr | Layered structures | Rhodamine B(RhB) | Xe lamp | 100 mL of RhB (5 ppm) | 1.0 | – | 99.8% photocatalytic degradation rate for RhB after 210 min | [99] |
| PDI/g-C$_3$N$_4$/TiO$_2$@Ti$_3$C$_2$ | Multi-layered 2D frame | Atrazine (ATZ) | 300 W Xe lamp (λ > 420 nm) | 50 mL of ATZ (10 ppm) | 0.8 | ~160 µmol·L$^{-1}$·h$^{-1}$ | 75% removal rate of ATZ within one hour | [100] |
| g-C$_3$N$_4$/α-MnS | Inhomogeneous morphology with a rough surface | Oxytetracycline (OTC) | 300 W Xe lamp (λ > 420 nm) | 50 mL of OTC hydrochloride (20 mg·L$^{-1}$) | 1.0 | 111.6 µmol·L$^{-1}$·h$^{-1}$ | 82.2% degradation of OTC in water within 80 min | [101] |
| Red mud/CdS | RM particles loaded on the surface of CdS nanospheres | Amoxicillin (AMX) | LED lamp (410 < λ < 760 nm) | 50 mL of AMX (20 mg·L$^{-1}$) | 0.5 | 1.05 mg·L$^{-1}$·h$^{-1}$ | 73.0% degradation of AMX within 120 min | [102] |

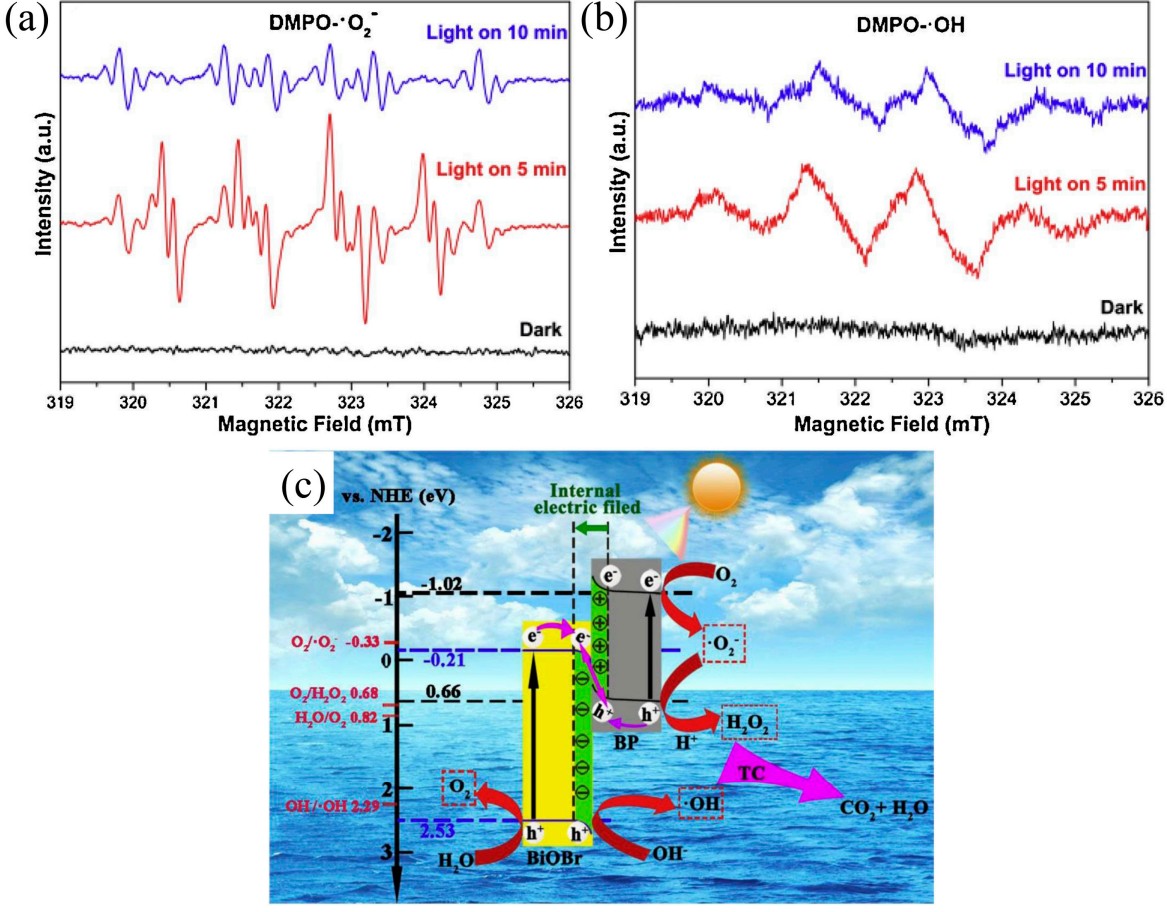

**Figure 10.** ESR under dark and visible light irradiation: (**a**) DMPO−·O$_2^-$ and (**b**) DMPO−·OH. (**c**) The mechanism of BP/BiOBr S-scheme photocatalyst [97].

Overall, the structure of S-scheme heterojunctions realizes rapid transfer and effective separation of photogenerated carriers and retains the strong redox capability of photocatalysts. This section shows the different S-scheme heterojunction photocatalysts in the literature for $H_2O_2$ production pathways and provides insights into the synthesis of efficient S-scheme heterojunction photocatalysts.

### 5. Conclusions and Outlook

Photocatalytic $H_2O_2$ production is a strategy used to avoid the drawbacks of conventional $H_2O_2$ production methods and, thus, achieve the conversion from solar energy to chemical energy. However, studies have shown that the efficiency and stability of single-component photocatalysts are not sufficient for practical applications. Therefore, modified photocatalysts obtained by constructing heterojunctions to facilitate the migration and separation of photogenerated carriers have been developed. The novel S-scheme heterojunction proposed by Yu's group overcomes the inherent defects of conventional heterojunctions and obtains a high redox capacity while promoting the effective separation of photogenerated carriers. This paper reviews the mechanism of novel S-scheme heterojunctions and photocatalytic $H_2O_2$ production and the application of S-scheme heterojunctions in the field of photocatalytic $H_2O_2$ production.

Up to now, the efficiency of photocatalytic $H_2O_2$ production has been limited by the energy band position of photocatalysts, the absorption ability of visible light and the migration and separation efficiency of photogenerated carriers. In particular, the inhibition of photogenerated carrier recombination is crucial for photocatalytic efficiency. It is shown that promoting the migration and separation of photogenerated carriers by constructing heterojunctions is most effective. In addition, there are two pathways for photocatalytic $H_2O_2$ production: two-electron ORR and two-electron WOR pathways. Most of the current studies have focused on the two-electron ORR pathway, which requires the addition of a hole sacrificial agent (isopropyl alcohol, ethanol, etc.) to facilitate the separation of photogenerated carriers. In contrast, the two-electron WOR pathway is rarely realized because it requires a higher oxidation potential than the four-electron WOR pathway to drive the reaction. Therefore, controlling the energy band structure to obtain a sufficient redox potential can improve the selectivity for $H_2O_2$.

S-scheme heterojunctions are found to be effective in enhancing visible light absorption, promoting the migration and separation of photogenerated charges, extending the lifetime of useful photogenerated charges and keeping a high redox capacity. However, the development of S-scheme heterojunctions in photocatalytic $H_2O_2$ production is still subject to various limitations. We propose the following aspects to promote the advancement of S-scheme heterojunctions in this field:

1.  Modification of the pore size, porosity and particle size of S-scheme heterojunction photocatalysts to increase their surface area, which is conducive to improving the adsorption of reactants ($H_2O$, $O_2$) by the photocatalysts;
2.  Construction of multiphase catalytic systems. At present, there are few studies on enhancing $H_2O_2$ yield by constructing multiphase S-scheme heterojunction photocatalytic systems. The disadvantage of slow gas transport kinetics of bi-phase catalysts can be avoided by constructing multiphase catalytic systems, which can promote the adsorption of $O_2$ by solid photocatalysts and further improve the efficiency of photocatalytic reactions;
3.  Combining photocatalysis with electrocatalysis. S-scheme heterojunctions are used to promote the separation of photogenerated charges by using intrinsic electric fields (IEF) at the interface, and other electric fields can be superimposed to further improve their separation efficiency. The introduction of an external electric field by applying a voltage can induce surface charge redistribution of the photocatalyst and can also facilitate the adsorption and activation of $O_2$ and $H_2O$;
4.  To construct the relationship between the Fermi energy level difference and redox potential. Modulation of redox potential by controlling the Fermi energy level positions

of semiconductors and constructing S-scheme heterojunctions to avoid four-electron competition reactions and improve the selectivity of $H_2O_2$ products;

5.  Optimize the model for theoretical calculations to pre-select semiconductors with suitable Fermi energy levels and energy band structures by theoretical calculations. Meanwhile, theoretical calculations combined with in situ characterization results can also enhance the investigation of the mechanism of photocatalytic $H_2O_2$ production and contribute to the deeper comprehension of interfacial charge transfer in S-scheme heterojunctions, which is important for the design of efficient S-scheme heterojunction photocatalysts;

6.  Considering future commercialization, in addition to the dual-channel pathway of photocatalytic $H_2O_2$ production, the cost of S-scheme photocatalysts should be controlled and recyclable and reusable photocatalysts should be designed.

Currently, the research of S-scheme heterojunctions in the field of photocatalytic $H_2O_2$ production is still in the preliminary stage. There are still many challenges on the road to commercialization of photocatalytic $H_2O_2$ production. We hope that our summary and outlook can facilitate the exploration of S-scheme heterojunctions in photocatalytic $H_2O_2$ production.

**Author Contributions:** Conceptualization, L.W. and W.F.; methodology, W.F.; software, W.F.; validation, L.W. and W.F.; formal analysis, W.F.; investigation, W.F.; resources, W.F.; data curation, W.F.; writing—original draft preparation, W.F.; writing—review and editing, L.W. and W.F.; visualization, L.W.; supervision, L.W.; project administration, L.W.; funding acquisition, L.W. All authors have read and agreed to the published version of the manuscript.

**Funding:** This research was funded by Fundamental Research Funds for the Central Universities, Ocean University of China (grant number 202364004).

**Data Availability Statement:** Not applicable.

**Conflicts of Interest:** The authors declare no conflict of interest.

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
