# Peer review of "S-Scheme Heterojunction Photocatalyst for Photocatalytic H2O2 Production: A Review"

_catalysts, doi:10.3390/catal13101325_

Round 1

Reviewer 1 Report

The work is well written and interesting.

One suggestion is to include more characterization of photocatalysts, so that other works can prepare more efficient materials.

Author Response

We appreciate the valuable comment from the reviewer. We have added the characterization of the photocatalysts in section 3.2 of the manuscript.

Reviewer 2 Report

Wang et al. summarized the design and application of S-scheme heterojunction for photocatalytic H2O2 production in this review. The basic principles and advantages of photocatalytic H2O2 production and S-scheme heterojunction are presented. This review is timely, and the paper is well organized. I would like to suggest the acceptance of this manuscript after minor revisions.

1. The sections “3.3. Synthesis Method” and “3.4. Photocatalytic Applications” should be deleted since this work is focused on the production of H2O2.

2. The subtitle of sections 4.1-4.4 should be revised to clarify the topic of this review. For example, the subtitle of 4.4 Pollutant degradation can be revised as Pollutant degradation coupled with H2O2 formation.

3. There are too many figures in this review and the number should be reduced to 8-10 based on the length of this review.

4. These references should be cited: ACS Appl. Mater. Interfaces 2021, 13, 6, 7238–7247; J. Mater. Chem. A, 2021,9, 1759-1769

No

Author Response

Wang et al. summarized the design and application of S-scheme heterojunction for photocatalytic H2O2 production in this review. The basic principles and advantages of photocatalytic H2O2 production and S-scheme heterojunction are presented. This review is timely, and the paper is well organized. I would like to suggest the acceptance of this manuscript after minor revisions.

  1. The sections “3.3. Synthesis Method” and “3.4. Photocatalytic Applications” should be deleted since this work is focused on the production of H2O2.

Response: We appreciate the valuable comment from the reviewer. We have modified manuscript. We've deleted section 3.4.

  1. The subtitle of sections 4.1-4.4 should be revised to clarify the topic of this review. For example, the subtitle of 4.4 Pollutant degradation can be revised as Pollutant degradation coupled with H2O2 formation.

Response: Thanks for your advice. We have modified the subtitles of 4.3 and 4.4 in manuscript.

  1. There are too many figures in this review and the number should be reduced to 8-10 based on the length of this review.

Response: Thanks for your valuable advice. We have modified manuscript.

  1. These references should be cited: ACS Appl. Mater. Interfaces 2021, 13, 6, 7238–7247; J. Mater. Chem. A, 2021,9, 1759-1769

Response: Thanks for your valuable advice. We have read the literatures carefully and have revised the manuscript according to them and have cited them in the manuscript.

Reviewer 3 Report

The authors describe the "S-scheme Heterojunction Photocatalyst for Photocatalytic H2O2 Production: A Review". The manuscript needs improvement in many ways. I recommend it be accepted for publication after major revision. The main concerns are as follows:

  1. Abstract: The main findings and important opinions are acceptable. The authors need to consider these points in the revision stage.
  2. Introduction: (i) The authors need to explain the effect of Z-scheme heterojunction photocatalyst on the H2O2 production using the recent references.

(ii) Please clarify the innovations of this work in the introduction section.

3.      In section 3.1: The authors need to include the mechanism of the Z-scheme heterojunction photocatalyst.

4.      In section 3.2: Need to add the characterization of Z-scheme heterojunction photocatalysts.

5.      In Figure 5: Provide the morphological analysis of the S-scheme heterojunction photocatalysts.

6.      In Figure 6: Add the photocatalyst details.

7.      In section 3.3: Better, the authors add the H2O2 production photocatalysts experimental details.

8.      In Table 1: (i) Most of the references belong to H2 production. So, the authors need to include the photocatalysts for H2O2 production applications.

(ii) Add the Z-scheme heterojunction photocatalysts and compare the H2O2 production performance with S-scheme heterojunction photocatalysts. 

9.      In Figure 8: Add the morphological analysis of the ZnO/WO3 photocatalyst.

10.  In Figure 9: Add the morphological analysis of the CdS/K2Ta2O6 photocatalyst.

11.  In Table 2: Include the morphology of the photocatalysts.

12.  In section 4.1: The authors need to discuss more about the summarized table 2 in the text.

13.  In Table 3: (i) Add the morphology of the photocatalysts.

(ii) Add more references related to S-scheme heterojunction photocatalysts.

14.  The authors need to compare it with Z-scheme heterojunction photocatalyst. How are S-scheme heterojunction photocatalysts more effective compared to Z-scheme heterojunction photocatalysts? Explain it.

15.  Are there any safety concerns or regulatory considerations related to the use of H2O2 production?

16.  How does this research contribute to our understanding of the fundamental science behind S-scheme heterojunction photocatalysts and their potential role in addressing energy and environmental challenges?

17.  What are the next steps in your research outline, and how do you envision the future of H2O2 production evolving based on these reports?

  1. The authors include their perspective on the future of this research in the conclusion section.
  2. The text is not free from grammatical, format, and punctuation errors. Please ask a native English speaker to revise and proofread their revised manuscript before re-submission.

Please ask a native English speaker to revise and proofread their revised manuscript before re-submission.

Author Response

The authors describe the "S-scheme Heterojunction Photocatalyst for Photocatalytic H2O2 Production: A Review". The manuscript needs improvement in many ways. I recommend it be accepted for publication after major revision. The main concerns are as follows:

Abstract: The main findings and important opinions are acceptable. The authors need to consider these points in the revision stage.

Introduction:

1.The authors need to explain the effect of Z-scheme heterojunction photocatalyst on the H2O2 production using the recent references.

Response: Thanks for your valuable advice. We have added recent references about Z-scheme heterojunction photocatalyst on the H2O2 production in introduction section.

2.Please clarify the innovations of this work in the introduction section.

Response: Thanks for your valuable advice. We’ve modified introduction section.

  1. In section 3.1: The authors need to include the mechanism of the Z-scheme heterojunction photocatalyst.

Response: Thanks for your valuable advice. We have added the mechanism of the Z-scheme heterojunction photocatalyst in section 3.1.

  1. In section 3.2: Need to add the characterization of Z-scheme heterojunction photocatalysts.

Response: Thanks for your advice. However, this review focuses on S-scheme heterojunction, and the title of section 3.2 is "Characterization of S-scheme Heterojunction", so we think that the characterization of Z-scheme heterojunction can be excluded from this section.

  1. In Figure 5: Provide the morphological analysis of the S-scheme heterojunction photocatalysts.

Response: Thanks for your valuable advice. We have provided the morphological analysis of the S-scheme heterojunction photocatalysts in section 3.2.

  1. In Figure 6: Add the photocatalyst details.

Response: Thanks for your advice. We have added the photocatalyst details in manuscript.

  1. In section 3.3: Better, the authors add the H2O2 production photocatalysts experimental details.

Response: Thanks for your valuable advice. We have modified section 3.3 in manuscript.

  1. In Table 1: (i) Most of the references belong to H2 production. So, the authors need to include the photocatalysts for H2O2 production applications.

(ii) Add the Z-scheme heterojunction photocatalysts and compare the H2O2 production performance with S-scheme heterojunction photocatalysts.

Response: Thanks for your valuable advice. We've deleted Table 1.

  1. In Figure 8: Add the morphological analysis of the ZnO/WO3 photocatalyst.

Response: Thanks for your valuable advice. We have added the morphological analysis of the ZnO/WO3 photocatalyst in Figure 6.

  1. In Figure 9: Add the morphological analysis of the CdS/K2Ta2O6 photocatalyst.

Response: Thanks for your valuable advice. We have added the morphological analysis of the CdS/K2Ta2O6 photocatalyst in Figure 7.

  1. In Table 2: Include the morphology of the photocatalysts.

Response: Thanks for your advice. We've modified Table 1.

  1. In section 4.1: The authors need to discuss more about the summarized table 2 in the text.

Response: Thanks for your valuable advice. We have modified manuscript.

  1. In Table 3: (i) Add the morphology of the photocatalysts.

(ii) Add more references related to S-scheme heterojunction photocatalysts.

Response: Thanks for your advice. We've modified Table 2.

  1. The authors need to compare it with Z-scheme heterojunction photocatalyst. How are S-scheme heterojunction photocatalysts more effective compared to Z-scheme heterojunction photocatalysts? Explain it.

Response: Response: Thanks for your valuable advice. We have modified in section 3.1 in manuscript.

  1. Are there any safety concerns or regulatory considerations related to the use of H2O2 production?

Response: The traditional method of H2O2 production is mainly the anthraquinone method, which produces a high concentration of H2O2 and has safety concerns during utilization. However, photocatalytic H2O2 production can produce low concentration H2O2 in situ, and there are no safety concerns in the process of use.

  1. How does this research contribute to our understanding of the fundamental science behind S-scheme heterojunction photocatalysts and their potential role in addressing energy and environmental challenges?

Response: Thanks for your valuable advice. This review summarizes the latest excellent research results of S-scheme heterojunction photocatalysts in the field of photocatalytic H2O2 production. The development of S-scheme heterojunction photocatalysts is understood from the perspectives of charge transport mechanisms, characterization methods, and preparation methods. S-scheme heterojunction photocatalytic system, not only the photogenerated electrons and holes can be effectively separated, but also still maintain the strong oxidizing and reducing properties of the original photocatalyst. On this basis, we hope to inspire researchers to construct more efficient and stable photocatalysts and to broaden the applications of S-scheme heterojunction photocatalysts in the field of photocatalysis, such as photocatalytic H2 production, CO2 reduction, and in-situ Н2O2 production to degrade pollutants.

  1. What are the next steps in your research outline, and how do you envision the future of H2O2 production evolving based on these reports?

Response: Thanks for the constructive suggestion. This review focuses on the mechanism of S-scheme heterojunction and its application in photocatalytic H2O2 production. However, the stability of photocatalyst decays during long-term photocatalysis. Therefore, the interactions of the S-scheme heterojunction system need to be further strengthened to maintain its long-term stability in the actual photocatalytic H2O2 production. Moreover, a lot of research has been conducted on the design and preparation of efficient S-scheme photocatalysts, but their catalytic efficiencies still cannot reach the industrial standards. Therefore, the photocatalytic performance of S-scheme heterojunction photocatalysts should be further improved.

  1. The authors include their perspective on the future of this research in the conclusion section.

Response: Thanks a lot for the reviewer’s carefully reading on the manuscript.

  1. The text is not free from grammatical, format, and punctuation errors. Please ask a native English speaker to revise and proofread their revised manuscript before re-submission.

Response: Thanks for the constructive suggestion. We have read the manuscript carefully and have made modifications.

Round 2

Reviewer 3 Report

The authors have responded to the reviewer's comments and can be accepted. 

Minor editing of English language is required during proof-reading.